# Recent Advances in Bioconjugate Vaccine Development

**DOI:** 10.3390/vaccines13070703

**Published:** 2025-06-28

**Authors:** Brendan W. Wren, Catherine L. Hall, Vanessa S. Terra, Mark A. Harrison, Elizabeth Atkins, Fauzy Nasher, Ian J. Passmore

**Affiliations:** Department of Infection Biology, London School of Hygiene and Tropical Medicine, Keppel St, London WC1E 7HT, UK; catherine.hall@lshtm.ac.uk (C.L.H.); vanessa.terra@lshtm.ac.uk (V.S.T.); mark.harrison1@lshtm.ac.uk (M.A.H.); elizabeth.atkins@lshtm.ac.uk (E.A.); fauzy.nasher1@lshtm.ac.uk (F.N.); ian.passmore@lshtm.ac.uk (I.J.P.)

**Keywords:** bacterial vaccines, glycoconjugate vaccines, carrier proteins, polysaccharides, protein glycan coupling technology (PGCT), bioconjugation

## Abstract

Glycoconjugate vaccines, consisting of a protein component covalently linked to a glycan antigen, have led to a significant reduction in the global occurrence of bacterial meningitis and pneumonia. They provide robust, lasting immunity in all age groups. However, their production by traditional chemical conjugation approaches has drawbacks in terms of complexity, cost, and lack of flexibility in design, which explains their limited application to a few pathogenic bacteria in the past four decades. Protein glycan coupling technology (PGCT) or bioconjugation, where glycoconjugates are produced in purpose-engineered bacterial cells, is a useful alternative to chemical conjugation and promises an array of low-cost custom-made glycoconjugate vaccines with vast protein glycan combinations. The technology has undergone significant development since its inception, and new advances and refinements continually drive the field forward. Several bioconjugate vaccines are currently in clinical trials, demonstrating the potential of the technology. We will review the wide applicability of bioconjugation and recent developments in each of the components of the technology, namely, glycan expression, protein selection, and the coupling of selected glycan with proteins, all within custom-designed *E. coli* cells. These advances promise to deliver effective glycoconjugate vaccines for multiple unmet medical needs.

## 1. Introduction

Vaccination is considered one of the great success stories of modern medicine and is an increasingly important strategy in the fight against antimicrobial resistance. Glycoconjugate-based vaccines are some of the safest and most efficacious vaccines in use today. Surface carbohydrates, such as capsular polysaccharides and O-antigens, have long been known to be ideal targets for bacterial vaccines.

Polysaccharide-only vaccines stimulate B cells to produce low-affinity, short-lived IgM antibodies but are unable to stimulate T cells for IgM to IgG switching and a sufficient memory response required for long-term protection [1]. By contrast, glycoconjugate vaccines interact with the immune system differently from polysaccharide-only vaccines and involve the processing of the peptide antigen in antigen-presenting cells and their presentation by major histocompatibility complex class II molecules. This subsequently activates CD4+ T helper cells to release interleukins that, in turn, initiate T cell-dependent B cell activation and antibody production [2,3]. Thus, glycoconjugate vaccines stimulate long-lasting antibody production and offer protection in all age groups, including children under 2 years of age.

The first glycoconjugate vaccine was designed against *Haemophilus influenzae* Type B and licensed in the 1980s [4]. The success of this vaccine led to the licensing of efficacious glycoconjugate vaccines, including those against *Neisseria meningitidis* [5], multiple serotypes of *Streptococcus pneumoniae* [6], and more recently, *Salmonella enterica* serovar Typhi [7]. Despite this success, the potential of glycoconjugate vaccinology is yet to be fully realized, and the development of vaccines against additional pathogens, including antibiotic-resistant bacteria, remains a global imperative.

The challenge of developing new glycoconjugate vaccines may lie in the current method of production: chemical conjugation, where glycans are traditionally harvested and extracted from the target bacteria in a multistep purification process. The glycan is then coupled to a separately purified protein carrier using chemistry that depends on the glycan’s functional groups. This process has significant drawbacks, such as multiple rounds of purification; batch-to-batch variation; and many quality control steps, hundreds in the case of the 13-valent pneumococcal vaccine Prevnar-13 (PCV13) [8]. These complications significantly increase the production cost of glycoconjugate vaccines, making some of them prohibitively expensive, which is of particular concern when considering they target infections with the highest burden in low- and middle-income (LMIC) settings. Nevertheless, many successful vaccines have been developed using this method [9]. Protein glycan coupling technology (PGCT), the enzymatic coupling of glycans to proteins in bacterial cells, has enormous potential in glycobiotechnology. When applied to the production of glycoconjugate vaccines, it is often termed bioconjugation. The major advantage of bioconjugation over chemical conjugation is that coupling occurs in vivo in a single step, avoiding the need for separate biosynthesis and purification of the carrier protein and glycan moieties. Bioconjugation, therefore, has the potential to yield an inexhaustible supply of convenient-to-produce glycoconjugate vaccines, reducing production cost and potentially increasing their accessibility to LMIC and veterinary vaccine markets, where low cost is paramount. Despite their promise, some barriers still exist for bioconjugate vaccines, which have limited their widespread development and use.

In this review, we will discuss recent strategies to overcome some of these barriers, such as yield limitation by optimizing each of the components used for synthesis, including the efficient expression of diverse glycans in *Escherichia coli* cells. We will demonstrate the wide applicability of the technology with the coupling of proteins to diverse glycans from important bacterial pathogens, including (i) O-antigens (e.g., *Francisella tularensis*), (ii) capsular polysaccharides (e.g., *S. pneumoniae*), (iii) *N*-linked glycosylation pathways (e.g., *Campylobacter* species), and (iv) conserved surface polysaccharides (e.g., Group A *Streptococci*).

## 2. Discovery of *N*-Linked Protein Glycosylation and the Development of PGCT and Bioconjugation

The first evidence of a dedicated general *N*-linked glycosylation system in bacteria was found in the gastrointestinal pathogen *Campylobacter jejuni* [10]. The genetic locus responsible was designated *pgl*, and its existence was later confirmed by genomic sequencing of the *C. jejuni* strain NCTC11168 in 2000 [11]. In this system, the glycan is covalently bound to the nitrogen atom of the amide group of an asparagine residue found within the consensus acceptor sequon D/EXNYS/T (where X and Y are any amino acid except proline) of a protein [12,13]. The most important finding from this *N*-linked glycosylation system was the discovery of the first example of a bacterial oligosaccharyltransferase (OST), PglB. In this seminal study, Wacker and colleagues demonstrated the functional transfer of the *pgl* locus into *E. coli*, which resulted in the heterologous biosynthesis of *C. jejuni* glycoproteins and opened the door for engineering of recombinant glycans and glycoconjugates for therapeutic applications [12]. In subsequent studies, *Cj*PglB was demonstrated to exhibit relaxed oligosaccharide substrate specificity, allowing it to transfer a diverse range of glycans from other bacteria to periplasmic proteins containing the acceptor sequon D/EXNYS/T [14,15]. Since then, *Cj*PglB has emerged as the most widely utilized enzyme for coupling glycans to acceptor proteins recombinantly in bacteria (discussed in detail below).

## 3. Recent Technical Development of Bioconjugation

To synthesize a bioconjugate vaccine, three components are typically required: (i) a glycan (e.g., glycosylation pathway, O-antigen, or capsular polysaccharide), (ii) a carrier protein, and (iii) a coupling enzyme. Each component is co-expressed and targeted to the periplasm in a bacterial cell, typically *E. coli* [16]. To date, efforts to improve and optimize each component have been treated as separate engineering challenges and have centred on three main goals: improved glycoconjugate yield through the genetic modification of the *E. coli* chassis and glycan biosynthesis pathways; expanding the range of glycans and acceptor proteins that can be coupled together through discovery and design of OSTs with alternative substrate specificities; and the development of pathogen-specific carrier proteins with superior immunological performance through reverse vaccinology (Figure 1).

### 3.1. Glycan

Bacteria biosynthesize a diverse variety of glycostructures, including capsular polysaccharides (CPSs), lipopolysaccharides (LPSs), and *O*- and *N*-linked glycans, the majority of which have not been definitively structurally characterized.

Many organisms display multiple glycans on their cell surface, which presents further challenges for vaccine design [17,18]. In the case of *Klebsiella pneumoniae*, there are over 140 capsular serotypes but as few as 13 O-antigen types [19]. In theory, an O-antigen-based vaccine would appear a more attractive option for designing a broad coverage multi-valent vaccine. However, evidence suggests capsule can physically block access of antibodies to the O-antigen, preventing their function and therefore decreasing vaccine efficacy. This suggests that capsule-based vaccines are superior to O-antigen-based vaccines in the case of *K. pneumoniae*, although the situation is likely to be different for other organisms. Matters are complicated further by serotype replacement, whereby non-vaccine serotypes occupy the niche vacated by serotypes that are present in the vaccine [20,21,22], which is a well-documented phenomenon for Streptococcal glycoconjugate vaccines. Flexible vaccine production platforms, such as those based on recombinant technologies, are likely to be important to keep pace with this shifting ecological landscape.

In bacteria, glycan biosynthesis pathways are frequently encoded in discrete operons that can be cloned and transferred en bloc to other host organisms such as *E. coli*. Heterologous biosynthesis of these glycans has been demonstrated for a number of clinically important bacterial pathogens, including capsular polysaccharide from Gram-positive and Gram-negative bacteria (e.g., *S. pneumoniae* [23] and *K. pneumoniae* [24]); lipopolysaccharides from Gram-positive and Gram-negative bacteria (e.g., *Streptococcus agalactiae*, *Francisella tularensis* [25], *Shigella dysenteriae*, *E. coli*, *K. pneumoniae*, and *Brucella abortus* [26,27,28,29,30,31]); conserved surface polysaccharides (e.g., Group A Streptococcus [32]) and general glycosylation pathways (e.g., *C. jejuni* [15]). However, as with any heterologous biosynthesis platform, faithful reconstitution of a functional pathway in a new host context can be challenging. Given the size and complexity of some glycan loci, yields can be lower than desired, and the degree of polymerization can be variable. The traditional approach has been to transfer an unmodified locus directly from the organism of interest into a suitable host, such as *E. coli*. However, this can be problematic if genes are not located in the same chromosomal location and cannot be amplified as a single contiguous DNA fragment. Furthermore, glycan biosynthesis operates within the specific regulatory and metabolic context of the native host. Transferring an unmodified pathway en bloc ignores this context and may present conflicts with endogenous sugar metabolism in the new host, ultimately leading to toxic effects or poor heterologous glycan yield. Recently, improvements have been made using synthetic biology and hierarchical combinatorial DNA assembly approaches [33,34,35].

Start–stop assembly is one such DNA assembly method that aims to build biosynthesis pathways with optimal configurations in a process known as ‘pathway refactoring’ [33]. In this approach, each coding sequence in the pathway is removed from its native regulation and paired with pools of standardized synthetic biology parts (such as promoters, ribosome binding sites, and transcriptional terminators) to vary transcription and translation of each gene. The pathway is then reassembled in modular format, generating pathway libraries that can be screened for optimal performance. The best-performing pathways can be selected and refined in subsequent iterative rounds of engineering. We recently demonstrated the merits of this approach by reconstructing biosynthesis pathways that faithfully produced the *C*. *jejuni N*-linked heptasaccharide [34]. Pathway clones were identified that outperformed the unmodified native *pgl* cluster, resulting in improved overall glycan and glycoconjugate yield.

In a comparable approach, we demonstrated the flexibility of combinatorial hierarchical assembly to biosynthesize the capsular polysaccharide of Group B Streptococcus serotypes III, IV, and V [35]. While the GBS CPS structures are antigenically distinct, their monosaccharide composition and genetic locus architecture exhibit remarkable similarities. We demonstrated that orthologous glycosyltransferases and metabolic enzymes could be used interchangeably between serotype pathways, which obviated the need to clone a new complete genetic locus for each CPS. Other modular DNA assembly techniques have also been developed for expressing non-native metabolic pathways in *E. coli*, including JUMP [36] and ePathOptimise [37].

In a similar vein, engineering of ‘scaffold glycans’, which contain core structures (common to multiple serotypes) that can be elaborated on with accessory monosaccharides and modifications, can be utilized to rapidly generate a diverse array of glycan structures. All O-antigen types of *S. flexneri*, except serotype 6, share a conserved polysaccharide backbone (corresponding to serotype Y) with the structure α-l-Rha*p*-(1→3)-α-l-Rha*p*-(1→3)-α-d-Glc*p*NAc. *S. flexneri* serotype diversity is found in modification of this backbone with glucosyl and O-acetyl groups by serotype-specific glycosyltransferases. Gasperini et al. recently demonstrated engineering of a serotype Y scaffold strain that could be supplemented with different O-antigen-modifying enzymes to generate 12 native serotypes and 16 novel serotypes that do not occur in nature [38]. While this platform was applied to the production of outer membrane vesicle vaccines, conceivably, this approach could be applied to the biosynthesis of recombinant glycoconjugate vaccines. Indeed, *S. flexneri* serotypes 2a and 3a glycans are substrates for *Cj*PglB and are included in the quadrivalent *Shigella* bioconjugate vaccine currently undergoing Phase II clinical trials (discussed below). These methods provide exciting opportunities for improving glycoconjugate production as well as expressing novel glycans for the development of glycoconjugates against a wider range of pathogens.

### 3.2. The Carrier Protein

The carrier protein of a glycoconjugate vaccine facilitates the presentation and recognition of the glycan antigen by T cells and the generation of immunological memory. Detoxified bacterial toxins, such as those from *Corynebacterium diphtheriae* (CRM197) and *Clostridium tetani*, are well-established carrier proteins used in licensed chemical glycoconjugate vaccines [39]. However, these proteins have not been utilized in the production of heterologous bioconjugates due to the requirement that they must be targeted to, and properly folded in, the periplasm. Instead, an alternative suite of acceptor proteins has emerged, which presents its own challenges and opportunities for custom vaccine design.

Most carrier proteins used in PGCT are not native glycoproteins and must be modified to enable their glycosylation, such as through the modification of the N-terminus with a periplasmic signal sequence and insertion of glycan attachment sites [26]. The sequence of glycosylation sites is OST-specific and can range from short amino acid motifs to longer peptides, depending on whether the attachment is an *N*- or *O*-linkage. *Cj*PglB recognizes the D/E-X-N-Y-S/T amino acid sequence, with asparagine as the site of glycan attachment [13]. *Desulfovibrio desulfuricans* also contain an orthologous *N*-linking OST with similarity to *Cj*PglB, but instead recognizes a shorter consensus sequence, N-X-S/T [40]. This recognition sequence is shared with the eukaryotic OST Stt3 and, as such, has been exploited for the development of humanized glycoproteins [41].

By contrast, *O*-linked OSTs typically glycosylate pilin-like proteins in their native host, at serine or threonine residues [42,43,44]. Unlike *N*-linked OSTs, these residues are not found within a defined amino acid motif but instead recognize protein structural conformations or amino acid content, such as regions with low complexity [45]. Minimal OST recognition sequences have been characterized for the pilin substrates of PilO, PglS, and PglL to facilitate their utilization in recombinant glycoconjugate vaccines, although these are generally longer than their *N*-linked counterparts, which limits the number that can be incorporated [46,47,48].

A carrier protein can be targeted for glycosylation by an OST by incorporation of its cognate recognition sequence into its amino acid primary structure. Such “glycotags” are typically fused to the termini of the protein or inserted within flexible protein loops, to ensure accessibility for the OST while preserving immunogenic peptide epitopes [49]. The most widely utilized carrier protein is a detoxified form of *Pseudomonas aeruginosa* Exotoxin A, which has been adopted in multiple bioconjugate vaccines, including those in Phase II/III clinical trials [50,51]. ExoA was originally engineered with two PglB glycosylation sites within flexible loops, but now, many iterations exist that incorporate additional glycotags at the N- and C-termini [25,26]. Similar strategies have also been employed towards the engineering of the acceptor proteins for *O*-linked glycoconjugates, toward the biosynthesis of vaccines against Group B Streptococcus and *Salmonella paratyphi A* [52,53].

Both the location of glycan attachment and glycan-to-protein ratio can influence glycoconjugate efficacy, and the ability to fine-tune these variables through carrier protein design is a distinct advantage of PGCT. The impact of glycan-to-protein ratio on immunogenicity is a complex relationship, influenced by glycan chain length, with optimal configurations specific to individual glycoconjugates [54,55]. In one example, the inclusion of an additional eight glycotags to the C-terminus of ExoA resulted in a protein that was hyperglycosylated with *F. tularensis* O-antigen compared with the 2-glycotag variant. This new formulation provides protection for rats against both the disease and the lethality associated with an inhalational challenge of *F. tularensis* Schu S4 [56]. Specifying the site of glycan attachment through glycotag positioning also offers the potential to enhance the immune response, through proximity to T cell and other immunogenic epitopes [2,57]. It is noteworthy that most studies investigating the impact of glycoconjugate configuration on immunogenicity are from chemical conjugates, where conjugation chemistry also plays a role; therefore, further research using PGCT-generated glycoconjugates is warranted.

Recombinant technologies such as PGCT offer inherent flexibility toward the design of bespoke carrier proteins and glycan combinations. Recently, increased interest has been placed in the development and application of novel carrier proteins with superior immune properties. Immune suppression, where repeated use of the same protein antigen (such as CRM197) in multiple vaccines leads to immunotolerance, is a recognized phenomenon that can blunt the efficacy of glycoconjugate vaccines as more are licensed [58]. Secondly, using both glycan and protein derived from the same pathogen, a so-called “double hit” approach, offers opportunities for enhanced immune stimulation. Third, the inclusion of a pan-species carrier protein could complement serotype-specific response limitations of immunizing with individual glycan antigens [59]. For example, Reglinski et al. used PglB to conjugate *S. pneumoniae* capsule 4 (SP4) to three conserved, immunogenic pneumococcal proteins (NanA, PiuA, and Sp0148), which demonstrated a similar anti-glycan response compared to commercial PCV, Prevnar-13, when all three conjugates were used in combination. This combination vaccine also generated antibodies that recognized a serotype 2 strain, whereas Prevnar-13 did not, demonstrating the potential advantages of using pathogen-specific carrier proteins [23]. However, immune responses varied when glycoconjugates were tested individually, demonstrating the need to investigate multiple carriers. More recently, NanA and a *Streptococcus pyogenes* protease, IdeS, were conjugated to RhaPS, a polyrhamnose homopolymer of the Streptococcal Group A Carbohydrate. Sera obtained following murine immunization found that compared to placebo, there was a significant IgG response to both NanA- and IdeS-RhaPS conjugates and a significant increase in IL-17A production, a marker of T cell activation [32].

The design and selection of carrier proteins can also be guided by principles of reverse vaccinology, a high-throughput, comprehensive method of using whole-genome sequencing data to assess all putative proteins within a pathogen’s proteome as novel antigens. A variety of computational tools are applied to assess a protein’s suitability as a vaccine candidate, such as conserved sequences, surface exposure, B and T cell epitopes, binding to immune cell receptors, toxicity, solubility, and structural considerations using AlphaFold. This is used to filter a subset of proteins of interest, which can then be experimentally validated, using a complementary high-throughput technique such as protein arrays [60,61,62]. More recently in silico antigen analysis has been enhanced through substantial developments in the understanding of the human B cell repertoire and protective epitopes, in combination with new bioinformatic methodologies [63,64]. Since its introduction in the early 2000s, when it was pioneered for the development of the meningococcal B vaccine, reverse vaccinology has been applied to an extensive list of human and animal pathogens [65,66,67,68,69]. This includes the identification of novel glycoconjugate carrier proteins, such as for the respiratory pathogen *Burkholderia pseudomallei* [70]. Thus, these proteins and any other proteins identified in a reverse vaccinology screen can be potentially coupled to a glycan for novel “double-hit” bioconjugate vaccines.

### 3.3. The Oligosaccharyltransferase Enzyme

Oligosaccharyltransferases (OSTs) are the cornerstone enzymes in bacterial protein glycosylation that mediate en bloc transfer of glycans from lipid-linked oligosaccharide (LLO) donors to specific amino acid residues in target substrate proteins. Since their discovery, these enzymes have been utilized for biotechnological ends toward the engineering of recombinant glycoproteins and glycoconjugate vaccines [71,72,73]. Bacterial OSTs modify either asparagine residues (*N*-linked) or serine/threonine residues (*O*-linked), each with varying sequon specificities and glycan substrate preferences.

#### 3.3.1. *N*-Linked OSTs

The prototypical bacterial *N*-linked OST is PglB from *C. jejuni* (*Cj*PglB), first identified as the key enzyme in the *C. jejuni N*-glycosylation pathway [10]. *Cj*PglB traverses the cytoplasmic membrane and catalyzes the transfer of preassembled glycans from an undecaprenyl-pyrophosphate (UndPP) donor to a conserved asparagine residue within the consensus sequon D/EXNXS/T in the periplasmic space [12]. Structurally and functionally homologous to the eukaryotic STT3 subunit, *Cj*PglB exhibits a broad glycan substrate tolerance, albeit with some limitations. For example, *Cj*PglB favors glycan substrates that contain an acetamido group at the C2 position of the reducing-end sugar, which narrows the spectrum of glycans that it can transfer. Structural-guided mutagenesis approaches have been deployed to engineer *Cj*PglB with broadened glycan substrate specificity, including toward the development of a glycoconjugate vaccine against *Salmonella enterica* Typhimurium, where the O antigen glycan end group has a galactose sugar that is incompatible with *Cj*PglB glycan coupling. Here, Ihssen et al. identified the glycan-interacting amino acid residues through three-dimensional homology models, which were then mutated to accommodate a galactose reducing-end sugar in the active site. Such studies have been aided by the high-resolution structural characterization of an orthologous PglB from the related species, *Campylobacter lari* [26,74].

One alternative approach to circumvent the substrate limitations of *Cj*PglB is through the modification of the glycan substrate. All strains of Group A Streptococcus display the immunogenic polyrhamnose Lancefield Group A Carbohydrate (GAC) on their surface, which makes it an appealing candidate for a universal vaccine. However, the native GAC glycan contains a reducing-end GlcNAc linked to rhamnose by a β-(1→4) linkage, which is not a substrate for *Cj*PglB [75,76]. Recently, Ajay-Castro et al. overcame the limitation of *Cj*PglB not recognizing β-1→4 GlcNAc at the reducing-end by engineering hybrid glycans with the modified reducing-end linkers of the glycan α-l-Rha*p*-(1→3)-α-l-Rha*p*-(1→3)-α-d-Glc*p*NAc and α-l-Rha*p*-(1→3)-α-l-Rha*p*-(1→2)-α-d-Gal*p*-(1→3)-β-d-Glc*p*NAc [32]. The linkers were assembled using biosynthesis pathways that included *Shigella dysenteriae* and *Shigella flexneri* O-antigen glycosyltransferases. The resulting glycoconjugate elicited carbohydrate-specific antibodies that bound the surface of multiple GAS strains, demonstrating the plasticity of the platform toward the construction of custom glycans.

#### 3.3.2. *O*-Linked OSTs

*O*-linking OSTs catalyze the transfer of glycans to serine or threonine residues on target proteins. Two broad mechanisms have been described: either en bloc transfer of a preassembled lipid-linked oligosaccharide or sequential *O*-glycosylation by glycosyltransferases that act directly on the protein. OSTs that transfer glycans en bloc have been most widely adopted for PGCT and have the major advantage that the enzymes are promiscuous in terms of transferring glycans with a range of end-groups.

En bloc *O*-glycosylation proceeds through a similar mechanism as *N*-glycosylation, whereby the glycan is first assembled on the UndP lipid carrier on the cytoplasmic face of the inner membrane, is flipped to the periplasm, and transferred to a target protein. A well-studied example is found in *Neisseria meningitidis*, where the OST (commonly referred to as *Nm*PglL) attaches an *O*-acetylated glycan to the type IV pilin protein, PilE [77,78], as well as a variety of other periplasmic and membrane proteins. In contrast to *N*-linked glycosylation, *Nm*PglL modifies target peptides which are found at flexible regions of low sequence complexity that are rich in small amino acids. While it accepts a broad range of glycan structures, PglL typically does not transfer polysaccharides with a glucose residue at the reducing-end and appears to favor substrates with N-acetylated sugars or galactose instead [79]. Despite the lack of a clear consensus sequence, efforts have been made to define an 8-amino-acid “minimal optimal *O*-linked recognition motif” (MOOR), analogous to the glycotags of *N*-linking enzymes. These recognition sequences have been incorporated into a range of protein carriers, including cholera toxin B, tetanus toxoid, and ExoA [47].

Several clinically important pathogenic bacteria have surface glycans that contain glucose at their reducing-end, including hypervirulent *Klebsiella pneumoniae* K1 and K2, all serotypes of *Streptococcus agalactiae*, and over 75% of *Streptococcus pneumoniae*. The soil microbe *Acinetobacter baylyi* encodes two *pglL* orthologues, one of which encodes an OST that modifies the pilin-like protein, ComP [44]. Crucially, this enzyme, denoted PglS, exhibits a notably broader glycan substrate specificity, including polysaccharides that contain a glucose residue at their reducing end, and therefore overcomes one of the main limitations of glycan substrate stringency of *Cj*PglB. Harding et al. utilized PglS to synthesize a polyvalent pneumococcal glycoconjugate vaccine by functionally transferring *pglS* and an *exoA-comP* fusion into *E. coli* [80]. In subsequent work, PglS has been utilized to generate multivalent conjugate vaccines against Group B Streptococcus and *K. pneumoniae* [18,24,31,52,78]. Like *Nm*PglL, efforts have been made to identify a PglS minimal peptide recognition sequence with a view to increasing the number of glycan attachment sites [48].

Other pilin-modifying *O*-OSTs have been identified, such as TfpO (also known as PilO), which was first identified in *P. aeruginosa* [81]. These typically modify the C-terminal serine of pilin proteins but are generally limited to transferring short oligosaccharides. In contrast, the more recently identified TfpM from *Moraxella osloensis* [82] shares sequence similarity with TfpO yet can transfer longer-chain polysaccharides, including those with glucose or galactose at the reducing-end.

### 3.4. Microbial Chassis Engineering

Engineering of heterologous glycoconjugates relies on the ability to coordinate the synthesis of the glycan antigen, OST, and acceptor protein under optimal conditions. Physical factors such as culture medium, growth temperature, induction conditions, and supplementation with cofactors and sugar substrates can be fine-tuned to optimize glycoconjugate yield [83]. Similarly, in a process known as chassis engineering, host cell metabolism can also be fine-tuned through the development of custom cells optimized for glycoconjugate production. Typically, this involves the targeted genetic modification of the host chromosome to remove conflicting factors that may drain the pool of available glycan substrates that can be directed toward glycoprotein synthesis. For example, deletion of the O-antigen ligase (*waaL*), which transfers UndP-linked glycan to the lipid-A core and competes with PglB for glycan substrate, results in markedly improved glycoconjugate yield [14]. Similarly, the cell availability of UndP can also influence the availability of glycan substrate for PglB. Maintaining higher levels of UndP can be achieved by deleting unessential phosphoglycerol transferases and glycosyltransferases and by increasing the rate of UndP synthesis by overexpressing the UndP(P)synthase, *uppS*. Kay et al. demonstrated that increasing the availability of UndP resulted in a 7-fold increase in biosynthesis of recombinant *S. pneumoniae* serotype 4 capsular polysaccharide [84]. Similarly, modifying metabolic flux, such as through the glyoxylate cycle [85], or blocking glycolysis and the pentose phosphate pathway [86], has also been shown to improve glycoconjugate yield.

Integrating components such as the OST *Cj*PglB onto the host chromosome can alleviate the metabolic cost associated with heterologous expression of a burdensome membrane-spanning enzyme [87,88]. Similarly, genome-editing strategies that involve the removal of the native enterobacterial common antigen (ECA) and *O*-polysaccharide antigen loci and their replacement with recombinant glycan biosynthesis pathways result in improved yields compared to when these components are provided on extrachromosomal plasmids [89].

The above examples highlight how a wide spectrum of different cellular processes can influence the synthesis of glycan structures. Forward metabolic engineering approaches that aim to rationally design cells with favorable properties are contingent on the availability of flexible genetic tools that can be used iteratively. Our group recently utilized CRISPR in combination with lambda red recombination to create a bespoke bank of *E. coli* strains specifically modified for heterologous glycan and glycoconjugate production [90]. Such mutagenesis methods introduce precise, scar-less deletions or insertions without the need to permanently insert antibiotic cassettes, which is particularly important for PGCT, where many components are supplied on plasmids that are maintained with antibiotics. Specifically, these strains aim to reduce contamination of glycoconjugate preparations through the deletion of *lpxM* and removal of endogenous endotoxin, increase monosaccharide availability through the introduction of exogenous glycan biosynthesis genes (such as the GlcNAc-GalNAc epimerase, *gne*) and increase polymer chain length by replacing the endogenous chain length regulator with that from the incoming pathway. These strains have been deposited at the Belgium coordinated collections of microorganisms and constitute an invaluable resource for the glycoengineering community. While these strains were specifically designed for the optimal production of *S. pneumoniae* serotype 4 CPS, they provide a blueprint that can be applied to other glycan and glycoconjugate synthesis strategies.

## 4. Selected Exemplars of Bioconjugate Vaccines with Different Glycan Structures

Despite some challenges, the field of bioconjugation has produced several successful prototype vaccines and vaccine candidates. Notably, *Shigella* and ExPEC *E. coli* vaccines, both developed using bioconjugation, are currently undergoing clinical trials and demonstrate the potential of this approach in vaccine development.

The first Phase I clinical trial involving a vaccine developed through bioconjugation [91] was conducted against *Shigella dysenteriae* by GlycoVaxyn. This study demonstrated that the vaccine was safe and able to elicit a specific and durable O1-antibody response. An important finding from this trial was that the carrier protein retained its immunogenicity even after bioconjugation [92], as seen previously for other targets [93].

Building on this, research continued toward a multivalent *Shigella* vaccine. A *Shigella flexneri* 2a vaccine, named Flexyn2a, was developed using the previously employed carrier protein, ExoA. Following this, a safety evaluation was conducted in which 30 healthy volunteers were administered 10 µg of polysaccharide bioconjugate. This formulation was again found to be safe and effective, with positive results from serum bactericidal assays and the detection of IgG and IgA antibodies [94]. Encouraged by these results, the vaccine was further tested using the controlled human infection model (CHIM) [51]. A Phase IIb trial was conducted at the Johns Hopkins Bloomberg School of Public Health, where 67 participants received two doses, one month apart, of either 10 µg Flexyn2a or a placebo. A month later, 59 participants were challenged with 1500 CFU of the *S. flexneri* 2a strain 2457T. The results showed that the vaccine provided 30% protection against moderate diarrhea and 72% protection against severe diarrhea. When vaccine efficacy was evaluated under more stringent conditions, such as fever or severe symptoms, vaccine efficacy was found to be 51.7%. This outcome is particularly promising in the context of CHIMs, as it better reflects the most serious symptoms of shigellosis [51]. Currently, clinical trials in Phase I/II are ongoing in Kenya, where a formulation is being tested as a quadrivalent vaccine targeting *S. flexneri* 2a, 3a, 6, and *S. sonnei* (S4V-EPA).

Another candidate currently in Phase I/IIa through Jansenn Pharmaceuticals is ExPEC10V, a vaccine designed to protect against various ExPEC O-serotypes (O1A, O2, O4, O6A, O8, O15, O16, O18A, O25B, and O75). As with the above example, the vaccine is conjugated to ExoA using PGCT. These serotypes were selected due to their significant association with bloodstream infections in the elderly across different geographic regions [50,95]. The vaccine has been shown to be both immunogenic and well tolerated in all adults enrolled in the study. An optimal dose range of 8–16 µg induced a robust increase in binding antibody titres for all serotypes tested. Notably, the immunogenic response remained above baseline throughout the first year. The generated antibodies demonstrated opsonophagocytic activity against all tested serotypes, except for O8. Based on these findings, a reformulated vaccine was developed, with O8 removed from the serotype panel. This change allowed for an increase in the O75 polysaccharide, which had shown the weakest immune response [95].

This quick redesign illustrates a key advantage of recombinant production platforms such as PGCT and their flexibility to adapt in real time based on emerging data. The updated formulation, ExPEC9V, is currently undergoing a Phase III clinical trial involving participants over 60 years of age with a history of urinary tract infections (UTIs) within the past two years.

## 5. Discussion and Future Perspectives

The use of bioconjugation as an alternative to chemical conjugation to produce glycoconjugate vaccines has come a long way in the past two decades, and it is now possible to make a multitude of prototype glycoconjugate vaccines quicker than they can be tested in animal infection models. The major advantages of the technology are the flexibility of design and the likely lower cost of vaccine product by using bioreactors to produce vaccines rather than traditional multistep chemical manufacturing processes that require large-scale equipment and multiple processing. The simplicity of the process facilitates “vaccine manufacturing for all” and the potential to produce vaccines in LMICs. This is timely with reduced funding for GAVI. Additionally, “do-it-yourself” bioconjugate vaccines can be custom-made for pathogens/serotypes that are important in local geographic regions. In addition, the flexible and low-cost production of bioconjugates will open new vaccine markets such as animal vaccines and maternal vaccines (protecting mother and child against Group B Streptococcus).

The immunological mechanisms of glycoconjugate vaccine protection are not fully understood [2,59]. Bioconjugation produces a vast range of glycoconjugate vaccine candidates, which, apart from vaccination, could be used to probe the immunological mechanisms of glycoconjugate vaccine protection in both humans and animals.

The yield of bioconjugates can be variable depending on the glycan, protein, and glycan–protein combination. The availability of a bank of *E. coli* host strains similarly will facilitate vaccine design, reduce toxicity, and production costs. Alternative host strains used in biotechnology, such as *Bacillus subtilis*, are being considered, which may be beneficial for the expression of components from Gram-positive pathogens and facilitating the production process, as products are directly secreted in the medium during growth.

The continual incremental gains in bioconjugate vaccines provide opportunities to produce the “more difficult to make vaccines” (e.g., a *Coxiella burnetii* bioconjugate vaccine). Similarly, the systematic expression of all 100+ *S. pneumoniae* capsular polysaccharide serotypes in different *E. coli* strains is underway to provide a lasting resource of clones that can be used for glycoconjugate vaccine design. Finally, in our continued fight to counteract antimicrobial resistance, bioconjugation offers the opportunity to target pathogens high on the WHO health emergencies list.

## Figures and Tables

**Figure 1 vaccines-13-00703-f001:**
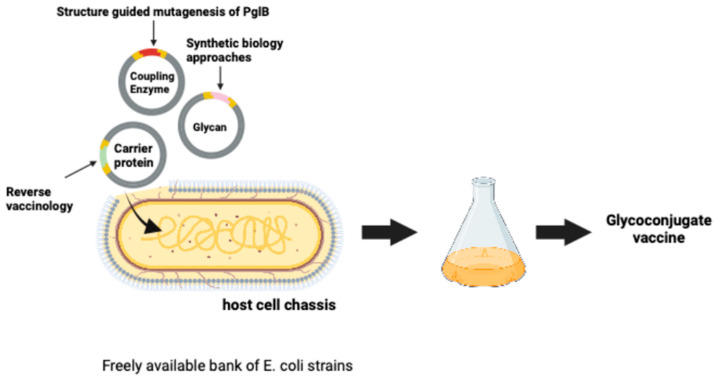
An engineering perspective of the three components of bioconjugation. Glycan yield improved using synthetic biology approaches, carrier protein choice improved by reverse vaccinology, and coupling of components improved using alternative OSTs and/or structure-guided mutagenesis. The chassis to express the vaccine components is a bank of custom-made *E. coli* cells.

## Data Availability

Data sharing is not applicable.

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
