# Peer review of "Recent Advances in Bioconjugate Vaccine Development"

_vaccines, 2025, doi:10.3390/vaccines13070703_

Round 1
Reviewer 1 Report
Comments and Suggestions for Authors
Very well written review particularly in the context of Bioconjugate vaccine development, but still some typo and update information are missing.
Introduction
Corrected “sero-types” to “serotypes”.
“barriers still exist for Bioconjugate vaccines, which have limited...” instead of “has limited.”
How can glycan expression be effectively controlled and diversified in engineered E. coli strains? - should be added in the introduction section
What specific criteria should guide the selection of protein carriers to ensure robust immunogenicity and compatibility with various glycans. Please add this section
To what extent can bioconjugation technologies meet global demand? explain in brief
One recent focus in bioconjugate vaccine development is the strategic consideration of glycan shielding, so it would be worthy to add this section in the introduction section
What are the key factors contributing to the high production cost of glycoconjugate vaccines, and how can these be minimized to improve accessibility in low- and middle-income countries (LMICs)?
Section 2
Line 90: “which resulted in the heterologous biosynthesis of C. jejuni glycoproteins, which opened the door” changed to “and opened the door” for better clarity.
Figure 1 should be more pictorial with high resolution image
Section 3.1
Why can transferring an unmodified glycan biosynthesis locus into E. coli be problematic?
What would be the role of post translation modification in bioconjugate.
"pathway refactoring" in the context of glycan biosynthesis, should be address in brief.
In what ways can ‘scaffold glycans’ streamline the development of glycoconjugate vaccines? Please elaborate in the revised manuscript
Section 3.2
Why traditional carrier proteins such as CRM197 not typically used in heterologous bioconjugate production. Input the detail about this
How can glycan-to-protein ratio and attachment site influence vaccine efficacy… this should be mentioned in the revised manuscript
In what ways has reverse vaccinology improved the identification of novel carrier proteins.. Suggest some reference.
Section 3.3
"oligosaccharlytransferase" in the title should be corrected to "oligosaccharyltransferase".
Input the primary function of oligosaccharyltransferases (OSTs) in bacterial protein glycosylation… and why glycosylation important for bacterial physiology and immune evasion.. Input couple of line to get familiar with reader to understand the importance.
Sub section 3.3.1
In the first paragraph, What is the consensus sequence recognized by CjPglB, and where in the cell does glycosylation occur... Have to mention in the revised manuscript.
Why was Salmonella enterica Typhimurium used as a model for engineering broader CjPglB specificity? Actually, looking for the advantages in the revised manuscript.
What innovative strategy did Ajay-Castro et al. use to modify the reducing-end sugar of GAC-like glycans? Elaborate it in detail, if possible.
Sub section 3.3.2
Line 343: and and.. Repeated.. Typo error
Line 348: “Other pilin-modifying O-OSTs has been identified…” should be have instead of has
Line 316-320: As suggested, two main mechanisms by which O-linked OSTs glycosylate proteins has been mentioned. Then further inform that OSTs that transfer glycans en bloc have been most widely adopted for PGCT.
Explain the major reason, why this is predominantly used in PGCT with major advantages ?
Why is PglS from Acinetobacter baylyi considered significant in O-linked glycoengineering? How did Harding et al. demonstrate the utility of PglS in glycoconjugate vaccine production? Elaborate it as the article suggest as referred ref. 80, if possible.
Section 3.4
Line 374: “Integrating components such as the OST CjpglB…”
"CjpglB" should be properly capitalized to match gene/enzyme naming conventions.
Replace with : “CjPglB”
Line 385: “…CRISPR in combination with lamda red recombination…” recheck the spelling of lamda.. It should be lambda
Line 398: last line “…glycan and glycoconjugates synthesis strategies…” should use singular to match context. Correct in the revised manuscript ( “…glycan and glycoconjugate synthesis strategies.”)
Input the function of the waaL gene, and why does its deletion enhance glycoprotein production. Explain it
Section 4
What was the first vaccine to undergo Phase I clinical trials using bioconjugation?
What is the S4V-EPA vaccine, and where is it being tested? Elaborate it !!
Section 5
Line 450 - 452: "Furthermore, the simplicity of the process facilitates “vaccine manufacturing for all” and the potential to produce vaccines in LMICs where they are most needed. - Slightly awkward phrasing, Please rewrite the sentence
Line 456-457: "Understanding of the expression of each component outlined in the figure will further improve yields and the choice of vaccines produced." - There is no figure in the conclusion. Either remove or clarify. It would be better to rewrite “...each component involved in the process…”
Author Response
We thank the reviewers for their useful comments and respond to queries in blue below. Changes in the text are highlighted in yellow.
Reviewer 1
Very well written review particularly in the context of Bioconjugate vaccine development, but still some typo and update information are missing.
Introduction
Correct “sero-types” to “serotypes”.
We agree that serotypes is the correct spelling, but the sero-types in the introduction is a split hyphen introduced at the end of the justified line by the journal edited script.
“barriers still exist for Bioconjugate vaccines, which have limited...” instead of “has limited.”
Corrected (line 73).
How can glycan expression be effectively controlled and diversified in engineered E. coli strains? - should be added in the introduction section
Sentence included (line 76).
What specific criteria should guide the selection of protein carriers to ensure robust immunogenicity and compatibility with various glycans. Please add this section
We list the criteria in lines 278 to 280, we have added the use of AlphaFold as a further tool to assess protein criteria.
To what extent can bioconjugation technologies meet global demand? explain in brief
This is a difficult question to answer and would require considerable speculation. We believe that partially answer this question in the discussion when we refer to “do it yourself vaccines” lines 476 to 478. We have modified this section to specifically mention the useful of Bioreactors compared to traditional more complex production approaches
One recent focus in bioconjugate vaccine development is the strategic consideration of glycan shielding, so it would be worthy to add this section in the introduction section
We agree that this is a good suggestion from the reviewer that may be used in the future for Bioconjugate vaccines. However, to date glycan shielding has only been used for viral based glycoprotein-based vaccines and is currently not attempted for the bacterial Bioconjugate-based approach in this review.
What are the key factors contributing to the high production cost of glycoconjugate vaccines, and how can these be minimized to improve accessibility in low- and middle-income countries (LMICs)?
We believe that this question is answered through lower production costs in the modified discussion lines 476 to 478.
Section 2
Line 90: “which resulted in the heterologous biosynthesis of C. jejuni glycoproteins, which opened the door”  changed to “and opened the door” for better clarity.
Corrected (line 94)
Figure 1 should be more pictorial with high resolution image
New Figure 1 added
Section 3.1
Why can transferring an unmodified glycan biosynthesis locus into E. coli be problematic?
We have modified this sentence (lines 151 to 153).
What would be the role of post translation modification in bioconjugate.
Post translational modification doesn’t occur in bacterial bioconjugates
"pathway refactoring" in the context of glycan biosynthesis, should be address in brief.
We have modified the sentence and explained the term "pathway refactoring" (line 157 to 167).
In what ways can ‘scaffold glycans’ streamline the development of glycoconjugate vaccines? Please elaborate in the revised manuscript
We have modified this section to make it more readable and have omitted the term scaffold glycans (lines 157 to 167).
Section 3.2
Why traditional carrier proteins such as CRM197 not typically used in heterologous bioconjugate production. Input the detail about this
CRM197 is not generally used in bioconjugate vaccines because of requirement for expression of carrier proteins in the periplasm and that alterative carrier proteins are required because of immune tolerance and the blunting of subsequent glycoconjugate vaccines that may have CRM197 as a carrier protein. This is explained in lines 202 to 204 and line 256.
How can glycan-to-protein ratio and attachment site influence vaccine efficacy… this should be mentioned in the revised manuscript
We have provided the example of the F. tularensis vaccine where the addition of 8 additional glycans per protein carrier increased protection from 50% to 100% in both the mouse and rat infections models (reference 56). This is modified in lines 242 to 246. Also highlighted is the attachment of the glycan and how that can influence efficacy in references 2 and 57 (lines 246 to 248).
In what ways has reverse vaccinology improved the identification of novel carrier proteins. Suggest some reference.
There are currently no specific published examples of where reverse vaccinology has improved the identification of novel carrier proteins for Bioconjugate vaccines. The references 66 to 70 show examples of proteins used in vaccines based on using the reverse vaccinology approach. We have included a sentence for clarification (lines 289 to 291)
Section 3.3
"oligosaccharlytransferase" in the title should be corrected to "oligosaccharyltransferase".
Corrected
Input the primary function of oligosaccharyltransferases (OSTs) in bacterial protein glycosylation… and why glycosylation important for bacterial physiology and immune evasion.. Input couple of line to get familiar with reader to understand the importance.
This sentence has been deleted as it is not particularly relevant to the main theme of the review.
Sub section 3.3.1
In the first paragraph, What is the consensus sequence recognized by CjPglB, and where in the cell does glycosylation occur... Have to mention in the revised manuscript.
We believe that this is covered in the text. The consensus sequon is D/EXNXS/T and glycosylation occurs in the periplasmic space.
Lines “CjPglB traverses the cytoplasmic membrane and catalyses the transfer of preassembled glycans from an undecaprenyl-pyrophosphate (UndPP) donor to a conserved asparagine residue within the consensus sequon D/EXNXS/T in the periplasmic space ​[12]​.”
Why was Salmonella enterica Typhimurium used as a model for engineering broader CjPglB specificity? Actually, looking for the advantages in the revised manuscript.
The sentence has been modified to specifically state the advantages for engineering CjPglB with broader specificity (lines 314 to 315).
What innovative strategy did Ajay-Castro et al. use to modify the reducing-end sugar of GAC-like glycans? Elaborate it in detail, if possible.
Sentences have been modified to specify the innovation of using linkers from other biosynthetic pathways to improve the compatibility of CjPglB in transferring a broader range of glycans (lines 325 to 326 and line 329).
Sub section 3.3.2
Line 343:  and and.. Repeated.. Typo error
Corrected
Line 348: “Other pilin-modifying O-OSTs has been identified…” should be have instead of has
Corrected
Line 316-320: As suggested, two main mechanisms by which O-linked OSTs glycosylate proteins has been mentioned. Then further inform that OSTs that transfer glycans en bloc have been most widely adopted for PGCT.
Explain the major reason, why this  is predominantly used in PGCT with major advantages    ?
A sentence has been added to specifically mention the major advantage of this class of OSTs in terms of recognising promiscuous end-group sugars (line 339 to 340).
Why is PglS from Acinetobacter baylyi considered significant in O-linked glycoengineering? How did Harding et al. demonstrate the utility of PglS in glycoconjugate vaccine production? Elaborate it as the article suggest as referred ref. 80, if possible.
 A sentence has been included to specify the advantage of Acinetobacter baylyi PglS (lines 360 to 363).
Section 3.4
Line 374:  “Integrating components such as the OST CjpglB…”
"CjpglB" should be properly capitalized to match gene/enzyme naming conventions.
Replace with : “CjPglB”
Corrected, we have adopted the convention of CjPglB for the naming of different OSTs through the text, ie The OST enzyme PglB that originated from C. jejuni.
Line 385: “…CRISPR in combination with lamda red recombination…” recheck the spelling of lamda.. It should be lambda
Corrected
Line 398: last line “…glycan and glycoconjugates synthesis strategies…” should use singular to match context. Correct in the revised manuscript ( “…glycan and glycoconjugate synthesis strategies.”)
Corrected
Input the function of the waaL gene, and why does its deletion enhance glycoprotein production. Explain it
Explanation provided (line 386 to 388)
Section 4
What was the first vaccine to undergo Phase I clinical trials using bioconjugation?
Sentence modified (line 430 to 431) highlighted in yellow.
What is the S4V-EPA vaccine, and where is it being tested? Elaborate it !!
Sentence modified (lines 450 to 452). It is difficult to expand on trials data undertaken by Pharmaceutical companies and we believe that this is not the theme of the review.
Section 5
Line 450 - 452: "Furthermore, the simplicity of the process facilitates “vaccine manufacturing for all” and the potential to produce vaccines in LMICs where they are most needed. - Slightly awkward phrasing, Please rewrite the sentence
Sentence modified, deleted last phrase (line 478 to 479)
Line 456-457: "Understanding of the expression of each component outlined in the figure will further improve yields and the choice of vaccines produced." - There is no figure in the conclusion. Either remove or clarify. It would be better to rewrite “...each component involved in the process…”
Sentence has been removed.
Reviewer 2
From a simple Medline search this the first overview focussing on the production and status of bioconjugate vaccines for over 5 years although some aspects are covered in more general reviews of glycoconjugate vaccines. It covers both generalities and some very specific considerations for bioconjugation and gives a good assessment of the state of the art. It provides a very valuable resource for experts to follow up.
The abbreviation of PGCT needs to be given when Protein Glycan Coupling Technology is mentioned in the text or perhaps better in the key words section.
The use of terms Protein Glycan Coupling Technology and PGCT have been checked throughout the text.
Reference 2 and 57 refer to the induction of anti-CHO T-cells with MHC binding glycopeptides. Has any advantage for this in vaccination been demonstrated? The review of Sun et al. 2016 <https://doi.org/10.1093/glycob/cww062> suggests that downregulating responses to pathogen-induced anti-CHO responses might be beneficial but perhaps there is more recent information.
To our knowledge there is no update on the mechanistic understanding of immune response to glycoconjugates. An advantage of bioconjugation is that a vast range glycoconjugates vaccine candidates can be produced with ease, which apart from vaccination, could be used to probe the immunological mechanisms of glycoconjugate vaccine protection in both humans and animals. We have included a sentence in the discussion to mention this point (lines 482 to 484).
The recent introduction of maternal vaccination there is a growing need for new carrier proteins to prevent maternal antibody inhibiting responses to new-born vaccination. Can the developments of bioconjugation be applied to problems with pathogens like S. pneumoniae?
We now mention the use of Bioconjugation for maternal vaccination in the discussion (lines 485 to 488).

Reviewer 2 Report
Comments and Suggestions for Authors
From a simple Medline search this the first overview focussing on the production and status of bioconjugate vaccines for over 5 years although some aspects are covered in more general reviews of glycoconjugate vaccines. It covers both generalities and some very specific considerations for bioconjugation and gives a good assessment of the state of the art. It provides a very valuable resource for experts to follow up.
The abbreviation of PGCT needs to be given when Protein Glycan Coupling Technology is mentioned in the text or perhaps better in the key words section.
Reference 2 and 57 refer to the induction of anti-CHO T-cells with MHC binding glycopeptides. Has any advantage for this in vaccination been demonstrated? The review of Sun et al. 2016 <https://doi.org/10.1093/glycob/cww062> suggests that downregulating responses to pathogen-induced anti-CHO responses might be beneficial but perhaps there is more recent information.
The recent introduction of maternal vaccination there is a growing need for new carrier proteins to prevent maternal antibody inhibiting responses to new-born vaccination. Can the developments of bioconjugation be applied to problems with pathogens like S. pneumoniae?
Author Response
ww
We thank the reviewers for their useful comments and respond to queries in blue below. Changes in the text are highlighted in yellow.
Reviewer 1
Very well written review particularly in the context of Bioconjugate vaccine development, but still some typo and update information are missing.
Introduction
Correct “sero-types” to “serotypes”.
We agree that serotypes is the correct spelling, but the sero-types in the introduction is a split hyphen introduced at the end of the justified line by the journal edited script.
“barriers still exist for Bioconjugate vaccines, which have limited...” instead of “has limited.”
Corrected (line 73).
How can glycan expression be effectively controlled and diversified in engineered E. coli strains? - should be added in the introduction section
Sentence included (line 76).
What specific criteria should guide the selection of protein carriers to ensure robust immunogenicity and compatibility with various glycans. Please add this section
We list the criteria in lines 278 to 280, we have added the use of AlphaFold as a further tool to assess protein criteria.
To what extent can bioconjugation technologies meet global demand? explain in brief
This is a difficult question to answer and would require considerable speculation. We believe that partially answer this question in the discussion when we refer to “do it yourself vaccines” lines 476 to 478. We have modified this section to specifically mention the useful of Bioreactors compared to traditional more complex production approaches
One recent focus in bioconjugate vaccine development is the strategic consideration of glycan shielding, so it would be worthy to add this section in the introduction section
We agree that this is a good suggestion from the reviewer that may be used in the future for Bioconjugate vaccines. However, to date glycan shielding has only been used for viral based glycoprotein-based vaccines and is currently not attempted for the bacterial Bioconjugate-based approach in this review.
What are the key factors contributing to the high production cost of glycoconjugate vaccines, and how can these be minimized to improve accessibility in low- and middle-income countries (LMICs)?
We believe that this question is answered through lower production costs in the modified discussion lines 476 to 478.
Section 2
Line 90: “which resulted in the heterologous biosynthesis of C. jejuni glycoproteins, which opened the door”  changed to “and opened the door” for better clarity.
Corrected (line 94)
Figure 1 should be more pictorial with high resolution image
New Figure 1 added
Section 3.1
Why can transferring an unmodified glycan biosynthesis locus into E. coli be problematic?
We have modified this sentence (lines 151 to 153).
What would be the role of post translation modification in bioconjugate.
Post translational modification doesn’t occur in bacterial bioconjugates
"pathway refactoring" in the context of glycan biosynthesis, should be address in brief.
We have modified the sentence and explained the term "pathway refactoring" (line 157 to 167).
In what ways can ‘scaffold glycans’ streamline the development of glycoconjugate vaccines? Please elaborate in the revised manuscript
We have modified this section to make it more readable and have omitted the term scaffold glycans (lines 157 to 167).
Section 3.2
Why traditional carrier proteins such as CRM197 not typically used in heterologous bioconjugate production. Input the detail about this
CRM197 is not generally used in bioconjugate vaccines because of requirement for expression of carrier proteins in the periplasm and that alterative carrier proteins are required because of immune tolerance and the blunting of subsequent glycoconjugate vaccines that may have CRM197 as a carrier protein. This is explained in lines 202 to 204 and line 256.
How can glycan-to-protein ratio and attachment site influence vaccine efficacy… this should be mentioned in the revised manuscript
We have provided the example of the F. tularensis vaccine where the addition of 8 additional glycans per protein carrier increased protection from 50% to 100% in both the mouse and rat infections models (reference 56). This is modified in lines 242 to 246. Also highlighted is the attachment of the glycan and how that can influence efficacy in references 2 and 57 (lines 246 to 248).
In what ways has reverse vaccinology improved the identification of novel carrier proteins. Suggest some reference.
There are currently no specific published examples of where reverse vaccinology has improved the identification of novel carrier proteins for Bioconjugate vaccines. The references 66 to 70 show examples of proteins used in vaccines based on using the reverse vaccinology approach. We have included a sentence for clarification (lines 289 to 291)
Section 3.3
"oligosaccharlytransferase" in the title should be corrected to "oligosaccharyltransferase".
Corrected
Input the primary function of oligosaccharyltransferases (OSTs) in bacterial protein glycosylation… and why glycosylation important for bacterial physiology and immune evasion.. Input couple of line to get familiar with reader to understand the importance.
This sentence has been deleted as it is not particularly relevant to the main theme of the review.
Sub section 3.3.1
In the first paragraph, What is the consensus sequence recognized by CjPglB, and where in the cell does glycosylation occur... Have to mention in the revised manuscript.
We believe that this is covered in the text. The consensus sequon is D/EXNXS/T and glycosylation occurs in the periplasmic space.
Lines “CjPglB traverses the cytoplasmic membrane and catalyses the transfer of preassembled glycans from an undecaprenyl-pyrophosphate (UndPP) donor to a conserved asparagine residue within the consensus sequon D/EXNXS/T in the periplasmic space ​[12]​.”
Why was Salmonella enterica Typhimurium used as a model for engineering broader CjPglB specificity? Actually, looking for the advantages in the revised manuscript.
The sentence has been modified to specifically state the advantages for engineering CjPglB with broader specificity (lines 314 to 315).
What innovative strategy did Ajay-Castro et al. use to modify the reducing-end sugar of GAC-like glycans? Elaborate it in detail, if possible.
Sentences have been modified to specify the innovation of using linkers from other biosynthetic pathways to improve the compatibility of CjPglB in transferring a broader range of glycans (lines 325 to 326 and line 329).
Sub section 3.3.2
Line 343:  and and.. Repeated.. Typo error
Corrected
Line 348: “Other pilin-modifying O-OSTs has been identified…” should be have instead of has
Corrected
Line 316-320: As suggested, two main mechanisms by which O-linked OSTs glycosylate proteins has been mentioned. Then further inform that OSTs that transfer glycans en bloc have been most widely adopted for PGCT.
Explain the major reason, why this  is predominantly used in PGCT with major advantages    ?
A sentence has been added to specifically mention the major advantage of this class of OSTs in terms of recognising promiscuous end-group sugars (line 339 to 340).
Why is PglS from Acinetobacter baylyi considered significant in O-linked glycoengineering? How did Harding et al. demonstrate the utility of PglS in glycoconjugate vaccine production? Elaborate it as the article suggest as referred ref. 80, if possible.
 A sentence has been included to specify the advantage of Acinetobacter baylyi PglS (lines 360 to 363).
Section 3.4
Line 374:  “Integrating components such as the OST CjpglB…”
"CjpglB" should be properly capitalized to match gene/enzyme naming conventions.
Replace with : “CjPglB”
Corrected, we have adopted the convention of CjPglB for the naming of different OSTs through the text, ie The OST enzyme PglB that originated from C. jejuni.
Line 385: “…CRISPR in combination with lamda red recombination…” recheck the spelling of lamda.. It should be lambda
Corrected
Line 398: last line “…glycan and glycoconjugates synthesis strategies…” should use singular to match context. Correct in the revised manuscript ( “…glycan and glycoconjugate synthesis strategies.”)
Corrected
Input the function of the waaL gene, and why does its deletion enhance glycoprotein production. Explain it
Explanation provided (line 386 to 388)
Section 4
What was the first vaccine to undergo Phase I clinical trials using bioconjugation?
Sentence modified (line 430 to 431) highlighted in yellow.
What is the S4V-EPA vaccine, and where is it being tested? Elaborate it !!
Sentence modified (lines 450 to 452). It is difficult to expand on trials data undertaken by Pharmaceutical companies and we believe that this is not the theme of the review.
Section 5
Line 450 - 452: "Furthermore, the simplicity of the process facilitates “vaccine manufacturing for all” and the potential to produce vaccines in LMICs where they are most needed. - Slightly awkward phrasing, Please rewrite the sentence
Sentence modified, deleted last phrase (line 478 to 479)
Line 456-457: "Understanding of the expression of each component outlined in the figure will further improve yields and the choice of vaccines produced." - There is no figure in the conclusion. Either remove or clarify. It would be better to rewrite “...each component involved in the process…”
Sentence has been removed.
Reviewer 2
From a simple Medline search this the first overview focussing on the production and status of bioconjugate vaccines for over 5 years although some aspects are covered in more general reviews of glycoconjugate vaccines. It covers both generalities and some very specific considerations for bioconjugation and gives a good assessment of the state of the art. It provides a very valuable resource for experts to follow up.
The abbreviation of PGCT needs to be given when Protein Glycan Coupling Technology is mentioned in the text or perhaps better in the key words section.
The use of terms Protein Glycan Coupling Technology and PGCT have been checked throughout the text.
Reference 2 and 57 refer to the induction of anti-CHO T-cells with MHC binding glycopeptides. Has any advantage for this in vaccination been demonstrated? The review of Sun et al. 2016 <https://doi.org/10.1093/glycob/cww062> suggests that downregulating responses to pathogen-induced anti-CHO responses might be beneficial but perhaps there is more recent information.
To our knowledge there is no update on the mechanistic understanding of immune response to glycoconjugates. An advantage of bioconjugation is that a vast range glycoconjugates vaccine candidates can be produced with ease, which apart from vaccination, could be used to probe the immunological mechanisms of glycoconjugate vaccine protection in both humans and animals. We have included a sentence in the discussion to mention this point (lines 482 to 484).
The recent introduction of maternal vaccination there is a growing need for new carrier proteins to prevent maternal antibody inhibiting responses to new-born vaccination. Can the developments of bioconjugation be applied to problems with pathogens like S. pneumoniae?
We now mention the use of Bioconjugation for maternal vaccination in the discussion (lines 485 to 488).
